# Synthesis of Nano Pigments Using Clay Minerals and Organic Dyes and Their Application as Colorants in Polymer Matrix

**DOI:** 10.3390/mi14051087

**Published:** 2023-05-21

**Authors:** Chandra Mohan, Neeraj Kumari, Philippe Jeandet, Priyanka Kumari, Arvind Negi

**Affiliations:** 1Department of Chemistry, School of Basic and Applied Sciences, K R Mangalam University, Gurugram 122103, India; 2Unit RIBP, USC INRAe 1488, University of Reims, 51100 Reims, France; 3Department of Chemistry, Shivaji College, University of Delhi, Delhi 110027, India; 4Department of Bioproduct and Biosystems, Aalto University, 02150 Espoo, Finland

**Keywords:** basic dyes, anionic dyes, nano pigments, polymeric films, textile

## Abstract

A new generation of clay-based nano pigments has been introduced, providing the advantage of both inorganic pigments and organic dyes. These nano pigments have been synthesized through a stepwise procedure where, initially, an organic dye is adsorbed onto the surface of the adsorbent, and then dye adsorbed adsorbent is used as pigment for further applications. The objective of the current paper was to examine the interaction of non-biodegradable toxic dyes, Crystal Violet (CV) and Indigo Carmine (IC), with clay minerals (montmorillonite (Mt), vermiculite (Vt), and clay bentonite (Bent)) and their organically modified forms (OMt, OBent, and OVt) and to develop a novel methodology for the synthesis of the value-added products and clay-based nano pigments without creating second generation waste materials. In our observation, the uptake of CV was more intense onto pristine Mt, Bent, and Vt, and the uptake of IC was more onto OMt, OBent, and OVt. CV was found to be in the interlayer region of Mt and Bent, as supported by XRD data. Zeta potential values confirmed the presence of CV on their surface. In contrast, in the case of Vt and organically modified forms, the dye was found on the surface, confirmed by XRD and zeta potential values. In the case of indigo carmine, the dye was found only on the surface of pristine Mt, Bent, Vt, and organo Mt, Bent, Vt. During the interaction of CV and IC with clay and organoclays, intense violet and blue-colored solid residues were obtained (also known as clay-based nano pigments). The nano pigments were used as colorants in a poly (methyl-methacrylate) (PMMA) polymer matrix to form transparent polymer films.

## 1. Introduction

Among various types of clay minerals, such as those of the smectite group, including montmorillonite (Mt), vermiculite (Vt), and clay, bentonite (Bent) is known to be highly reactive because of their larger surface area and high cation exchangeability, increased swelling, and adsorption capacity [1,2,3,4,5]. Clay minerals interact with various organic/inorganic compounds, such as dyes, drugs, and pigments, in different ways. The idea of the interaction of organic dyes with inorganic clay minerals is not new, as it is already known that clay interacts with organic dyes through electrostatic interaction, secondary bonding, and covalent bonding to produce valuable products (dye/clay hybrid nano pigments) that have several uses in the manufacturing industries [6,7]. Most research has focused on extracting dyes from aqueous media under various constraints such as pH, contact time, initial concentration, adsorbent dosage, ionic strength, and temperature using batch extraction studies. Still, no reports show the formation of nano pigments through the interaction of organic dyes with clay minerals by modulating various parameters [7].

Nano pigments are hybrid materials formed by combining organic dyes with silicate clays. The clay particles immobilize the dye molecules, improving dyes’ physicochemical properties, such as color strength and thermostability [8]. Jaber et al. synthesized hybrid pigments based on carminic acid and Montmorillonite clay minerals [9]. Hybrid dye-clay nano pigments were synthesized using Methylene blue and Cloisite 15A through the ion exchange method and used further for organic coating [10]. Osajima et al. modified bentonite clay using cetyltrimethylammonium bromide as a reinforcing agent for the photostabilization of anthocyanin dye [11].

In recent decades, studies reported the synthesis of nano pigments using cation exchange methods (organic dyes intercalated in smectites focusing on improving the photochemistry of intercalated dyes with the dispersion of clay minerals) [12,13,14,15]. However, no reports on synthesizing nano pigments using the batch extraction method have been disclosed until this date [9,10]. Therefore, the dyes chosen for the present study are nonbiodegradable dyes that have higher cellular toxicity (Crystal Violet (CV) and Indigo Carmine (IC)) (as shown in Figure 1). CV has a triaryl methane as a chromophoric group belonging to the basic dyes category. In contrast, IC belongs to the anionic dyes category consisting of a cross-conjugated system with C=C double bonds substituted by two N-H donor groups and two C=O acceptor groups [16,17,18].

CV is used in the textile industry as a dyeing colorant, whereas IC is used on a large scale to produce denim cloths for blue jeans and other denim products. However, their (CV and IC) exposure can permanently injure the cornea, conjunctiva, and skin, eye irritation, vomiting, and diarrhea in humans [19].

The foremost aim of the current paper was to develop a novel methodology for the synthesis of clay-based nano pigments without creating second-generation waste products and to use them in polymer matrixes as colorants to produce bright-colored polymeric films having high thermal and UV stability with enhanced mechanical properties, as compared to pure polymeric films and polymeric films containing pure dye.

## 2. Materials and Methods

### 2.1. Materials

The reagents utilized during experimental work were of analytical grade. All the naturally occurring clay minerals (montmorillonite and vermiculite), clay (bentonite), and Crystal Violet were procured from Sigma Aldrich Chemical Pvt. Ltd., St. Louis, MO, (USA), and Indigo Carmine was procured from Thomas Baker Chemical Ltd. (Mumbai, India). Cetylpyridinium chloride (CPC) was procured from Merck Pvt. Ltd. (Mumbai, India), and Poly (methyl-methacrylate) polymer was procured from Taj Resins Pvt. Ltd. (New Delhi, India). All other compounds were of analytical quality and were utilized as-is in this research.

### 2.2. Synthesis of Organoclay

Organoclay has been prepared through ion exchange processes by modifying a reported procedure [20]. For the synthesis of organoclays, the pristine clay minerals were treated with dissimilar concentrations of CPC (2% of CPC).

A known amount (5 g) of the naturally occurring clay/clay minerals (Mt/Vt/Bent) was added in 400 mL of double distilled water and left for stirring for 24 h. After 5–6 h of stirring, 100 mL of 2% CPC in clay suspension was added to the clay dispersion, and the mixture was kept on the stirrer for another 2 h. Later, the solution was left to settle down. Centrifugation was performed at 8000 rpm for 20 min to separate the residue from the supernatant. Dust was obtained by air-drying and crushing the remnants using a pestle and mortar. Organoclays, OMt, OBent, and OVt are the names given to these residues.

### 2.3. Physicochemical Characterization of Organoclay and Polymer Films

X-ray diffraction (XRD) patterns were investigated through a powder X-ray diffractometer (Philips X′ Pert-PRO Panalytical, model 3040160) using Cu Kα line (λ = 1.54056 Å) having 2θ values between 2° to 30°. The samples’ zeta potential and particle size were recorded using a Malvern zeta sizer (Ver. 6.01). Pure PMMA and PMMA films with different inorganic fillers were tested for their tensile strength, and young’s modulus (mechanical characteristics; dimensions: 50 mm × 15 mm × thickness: 0 mm to 23 mm) was recorded using a tensile testing machine (Instron UTM 3369). The tensile strength of PMMA films was calculated using the following formula:(1)$$\mathrm{Tensile}\;\mathrm{strength}=\frac{\mathrm{Force}\;\left(\mathrm{maxiumum}\;\mathrm{load}\;\mathrm{applied}\;\mathrm{to}\;\mathrm{elongate}\;\mathrm{the}\;\mathrm{film}\right)}{\mathrm{Area}\;\left(\mathrm{width}\;\times \mathrm{thickness}\right)}$$

PMMA and PMMA films with various inorganic fillers were examined for their hardness using a Shore A Durometer, which measures the depth of a part of this procedure in a specimen by applying a force to an indenter foot held at a right angle to the surface of the specimen. The light fastness study of colored PMMA films was done using a xenon arc lamp (ASTM G 155-08), MICOM Laboratories INC, Canada, for 100 h. The standard and the light source distance was 6 ± 1 cm under 62 °C temperature and 50% relative humidity. The value of ΔE was measured before and after light irradiance using the CIE lab difference equation:(2)ΔE=(ΔL*2+Δa*2+Δb*2)12
where:
ΔE is the color difference between the standard sample and the tested sampleΔL* represents the white-black axisΔa* is the red-green axisΔb* is the yellow-blue axis


The light fastness study was done by taking two samples of the polymer films:PMMA film containing the pure dyePMMA film containing the organo clay-based nano pigments

The standard sample was kept in the dark for the light fastness study, and the tested sample was under UV radiation.

### 2.4. Methodology for Batch Extraction of CV and IC Using Mt, Bent, and Vt

The interaction of each dye (CV and IC) with pristine and organo form of Mt, Bent, and Vt was investigated in aqueous solutions using the batch extraction mode as a function of:The pH of the aqueous medium of dyeThe contact time of batch withdrawalThe initial concentration of dye

The stock solution of CV and IC dye (3000 mg L^−1^) was prepared in double distilled water and diluted to achieve the required concentration using double distilled water. For each batch extraction experiment, the amount of pristine and organoclays (0.1 g) and the volume of the aqueous solution of CV and IC (25 mL of the total volume of an aqueous solution of dye) of a concentration of 50 mg L^−1^ (except the studies conducted as a function of the initial concentration of dye) was kept constant. At the end of each experiment, the sample was centrifugation for 20 min at 8000 rpm using Remi, R-24 centrifuged machine, and concentrations of CV and IC in the supernatants were estimated using the UV-Visible spectrometric method.

The following relation was used for the estimation of the amount of CV and IC retained on the pristine and organo Mt, Bent, and Vt:(3)$$q_{e}=\frac{C_{i}-{\;C}_{e}\times V}{m}$$
where:
q_e_ (mg g^−1^) is the amount of CV/IC retained by pristine and organo Mt, Bent, and Vt.C_i_ (mg L^−1^) is the aqueous solution’s initial concentration of CV/IC.C_e_ (mg L^−1^) is the aqueous solution’s equilibrium concentration of CV/IC.V (L) is the volume of an aqueous solution of CV/IC.m (g) is the pristine and organo Mt, Bent and Vt used.


## 3. Results and Discussion

### 3.1. XRD Pattern of Clay before and after Modification

XRD pattern of clay before and after modification with CPC is shown in Figure 2. XRD pattern of Mt and Bent showed the characteristics diffraction angle corresponding to 001 plane at 2θ value of 6.2° and 5.9° with the basal spacing 14.25 Å and 15.09 Å. On interaction with CPC, the diffraction angle of pristine Mt and Bent shifted toward lower angle having 2θ value 5.3° and 5.4° resulting increase in basal spacing of Mt (17.0 Å) and Bent (16.38 Å). The increase in the basal spacing was due to the intercalation of CPC in the interlayer of Mt.

The characteristics diffraction angle of Vt was at 2θ value of 6.0° showing the interlayer spacing 14.72 Å. On interaction with CPC, there was no significant change in diffraction angle of Vt suggesting no intercalation of CPC in the interlayer of Vt. The possibility of interaction of CPC was found with the surface.

The same interlayer spacing of Vt after interaction with CPC showed the limited expansion of the interlayer of Vt as compared to pristine Mt and Bent suggesting the negligible ion exchange capacity of pristine Vt [21].

### 3.2. FT-IR Spectra of Clay before and after Modification

FT-IR spectrum of clay before and after modification with CPC is shown in Figure 3. FT-IR spectrum of Mt and Bent showed the vibrational bands at 3624 cm^−1^, 3618 cm^−1^, 3410 cm^−1^, and 3427 cm^−1^ due to O-H stretching vibration of structural O-H group, H-O-H stretching vibration of interlayer water in Mt and Bent. FT-IR spectrum of Vt showed vibrational bands at 3400 cm^−1^ due to O-H stretching vibration of water present in the interlayer and available silanol group bonded to the surface. The vibrational bands in Mt, Bent and Vt at 1648 cm^−1^ and 1640 cm^−1^ have been assigned to the H-O-H bending vibration of surface and/or interlayer water. The stretching vibration of Si-O-Si and Si-O-Al for Mt, Bent, and Vt is observed as a very strong broad absorption band at 1043 cm^−1^, 1020 cm^−1^, and 996 cm^−1^.

On interaction with CPC, the FTIR spectrum of OMt, OBent, and OVt showed a new pair of the vibrational bands at 2929 cm^−1^, 2858 cm^−1^ in OMt, at 2925 cm^−1^, 2858 cm^−1^ in OBent, and at 2917 cm^−1^ and 2851 cm^−1^ in OVt which has been assigned to the asymmetric and symmetric vibrations of methylene group of the CPC present on the surface of the OMt, OBent, and OVt. Furthermore, a substantial reduction in the intensity of H-O-H stretching vibration of interlayer water at 3410 cm^−1^ in Mt, at 1648 cm^−1^, 1640 cm^−1^ in Mt, Bent further confirms the replacement of most of the interlayer water by CPC and thus ensures the presence of CPC on the surface as well as in the interlayer region, also supported by XRD data [22].

### 3.3. Effect of pH on the Stability of CV and IC in Aqueous Media

The electronic absorption spectrum of CV in an aqueous solution shows four main absorption bands: 210, 251, 305, and 594 nm at its natural pH of 5.5 (Figure 4A). The absorption band at 210 and 251 nm is due to π-π* transition of the aromatic ring present in CV. The chromophoric group of CV, triphenylmethane, is mainly responsible for the absorption band at 305 and 594 nm [23].

The behavior of the CV dye molecule as a function of pH was evaluated by recording CV’s electronic absorption spectrum of aqueous solutions. It was found that the λ_max_ value (594 nm) of aqueous solutions of CV remains unchanged in a pH range of 3 to 9. However, their absorbance values were influenced by the pH of the solution (Figure 4B). At pH 1, the color of the aqueous solution of CV was yellow and became colorless due to the protonation of all three nitrogen atoms present in CV. In contrast, at pH 2, the color was green with different λ_max_ values, 420 and 630 nm, due to the protonation of two of the nitrogen atoms, which disturbs the conjugation in the structure of CV. At pH 10, the aqueous solution of CV became colorless due to the formation of the carbinol form of CV [5]. However, the value of the extinction coefficient was maximum at pH 4. As a result, pH 4 was selected for all the quantitative estimations of CV. In an aqueous solution, CV was found to follow Beer–Lambert’s law within a concentration range of 0.5 mg L^−1^ to 4.5 mg L^−1^ with a correlation coefficient value of 0.9999 (Figure 4C). The electronic absorption spectrum of IC shows four absorption bands located at 205, 253, 289, and 612 nm at its natural pH, 4.2 (Figure 5A). The absorption band located at 612 nm is due to the chromophoric indigo group present in IC [24].

The absorption spectra of IC in an aqueous solution were tested across a wide range of pH values (1–10) and were unaffected by the solution’s acidity (Figure 5B). Because of this, the quantitative estimate of IC was performed at pH 4 (near the natural pH of the aqueous solution of IC). The IC concentration–response curve in water was linear between 5 mg L^−1^ and 55 mg L^−1^, with a correlation coefficient of 0.9995 (Figure 5C).

### 3.4. Interactions of CV with Pristine Clays and Organoclays

Various characteristic properties of pristine and organo Mt, Bent, and Vt (treated with 0.5 % CPC) and their respective CV-treated forms are presented in Table 1. The uptake of CV was found to be substantially high in pristine Mt, Bent, and Vt due to the strong electrostatic interaction. After interaction with CV, the surface charge of Mt, Bent, and Vt gets changed to −9.42 mV, +35.5 mV, and −14.6 mV from −17.6 mV, −31.1 mV, and −47.0 mV respectively, indicating the presence of CV on their surface. An increase in the particle size of CV-treated pristine clays further supports the surface retention of CV. From XRD data, it can be said that CV is intercalated between the interlayer region of Mt and Bent. However, in the case of Vt, CV interaction was found to be limited only to the surface, which might be due to Vt’s negligible ion exchangeability (Figure 6 and Figure 7).

Upon CPC (0.5%) interaction with pristine clays, intercalated and surface-modified species were formed. In the case of Mt, Bent, and Vt, XRD data confirmed complete intercalation with surface interactions and only surface interactions, further supported by zeta potential values and particle size data. The modification of the surface charges of pristine clays was due to the single layer of CPC.

The uptake of CV was found to be less in the case of organoclays than in pristine clays. Since CPC occupied the interlayer spaces of OMt and OBent, the possibility of interactions of CV was found with available negative sites on the surface of organoclays resulting in further reduction of negative charges as further supported by the particle size and XRD data (Table 1). The interlayer spacing of OMt decreased after interaction with CV, either due to the replacement of CPC by CV from the interlayer spaces of OMt or surface interactions.

### 3.5. Interaction of IC with Pristine Clays and Organoclays

Various characteristics and properties of pristine and organoclays (treated with 2% CPC) and their respective IC-treated forms are presented in Table 2. Since IC is anionic and pristine clays possess a negative charge. Thus, it was revealed that IC might interact with the positive edges of pure clays (Figure 8). However, the particle size data further confirms that IC only interacts with the surface of pure clays since neither the surface charge nor the interlayer spacing changed much after exposure to IC.

With 2% CPC, the formation of double layers of CPC on the surface of organo Mt and Bent increased the positive charge density. The surface charges of Mt and Bent got modified to +29.3 mV and +35.5 mV from −17.6 and −31.1, whereas the surface charges of Vt remained constant, indicating the saturation of the surface of Vt with 0.5% CPC. As a result, the 2% surfactant did not cause further changes in the surface charge density (Table 2).

Due to the strong electrostatic interaction between the positively charged surface of organo Mt, Bent (treated with 2% CPC) and the anionic IC, uptake of IC was observed to be substantially high (Figure 9). Upon interaction with IC, the surface charges of OMt and OBent got changed to −2.61 mV and −13.4 mV from +29.3 mV and +35.5 mV, respectively, indicating the presence of IC on the surface of OMt and OBent. IC being anionic, the sulfite group of IC interacts with the positive pyridinium group of CPC at the surface of organoclays. XRD and particle size data of IC-treated organo clays further supported the surface retention of IC. Since the surface charge of OVt remained negative even after treatment with 2.0% CPC, the uptake of IC was less compared to OMt and OBent.

### 3.6. Synthesis of Clay-Based CV/IC Nano Pigments

During the extraction of CV and IC from aqueous media, beautiful violet and blue-colored residues were obtained as clay-based nano pigments. Clay-based nano pigments were synthesized using the optimized conditions obtained during the extraction of CV and IC from aqueous media. Ten grams of pristine clays/or organoclays were added to 250 mL volumes of an aqueous solution of dyes (CV/IC) at the optimized pH. The content was magnetically stirred for the optimized time. Afterward, the mixture was centrifuged for 20 min to separate pigments from the supernatant. The clay-based pigments thus obtained were dried at 80 °C in an oven overnight. These pigments were mechanically crushed to form fine particles using a pestle and mortar and used for further studies Figure 10.

### 3.7. Application of Clay-Based CV/IC Nano Pigments

#### 3.7.1. Role of Clay-Based Nano Pigments in Polymer Matrix

Clay-based nano pigments are a class of composite materials with the advantages of organic dyes and inorganic pigments, synthesized by combining organic dyes and layered silicates. The major applications of clay-based nano pigments are in polymeric materials, which can be used as coloring agents in the polymer matrix to form thin, transparent, and beautifully colored polymer films [25]. Nano pigments play a dual role in the polymer matrix; they provide color to the polymer matrix and act as inorganic fillers by enhancing their physicochemical properties without accumulating second-generation-waste materials [26]. However, as pristine clays are hydrophilic, they do not show good compatibility with polymer matrixes resulting in the visibility of individual particles due to their poor dispersion. Therefore, for better compatibility and homogenous dispersion, the clay surface needs to be modified using a cationic surfactant [27]. To obtain bright-colored polymer films, clay-based nano pigments should be brightly colored. Hence, the interaction of the cationic dye, CV, was more with pristine clays than organo clays. The latter did not show good compatibility with the polymer matrix due to the pristine clays’ hydrophilic nature, resulting in visible individual particles in the polymer film [28]. After modification of the surface of the pristine clay, the uptake of CV decreased due to the neutralization of the surface charges of pristine clays. As the concentration of CPC increased from 0.5% to 2%, the positively charged surface of the organoclays obtained resulted in light-colored residues due to poor uptake of CV. Therefore, the interaction of CV was shown only with 0.5% CPC-treated organoclays. IC, anionic in nature, displayed more interaction with the positively charged surface of organoclays; therefore, IC uptake increased with increasing CPC concentration, from 0.5% to 2%.

#### 3.7.2. Application in PMMA Films Preparation Using Inorganic Fillers

The dispersion of various inorganic fillers prepared a series of PMMA polymer films in PMMA polymer matrixes by the solvent casting method:Pristine clay/clay minerals (Mt, Vt, Bent)Organo clays (OMt, OBent, OVt)Crystal Violet/Indigo Carmine dyePristine clay minerals-based CV/IC nano pigmentsOrgano clay-based CV/IC nano pigments

A total of 5 g of Poly (methyl methacrylate) (PMMA) was mixed with 25 mL of tetrahydrofuran (THF) solvents in a 100 (mL) beaker to create a solvent for the synthesis of PMMA film comprising diverse components. The polymer was entirely dissolved after 5–6 h of stirring. Maintaining the stirring (for about an hour) in the polymer solution, 3% (150 mg) of the inorganic filler was added until full dispersion of the resulting inorganic filler was attained. After allowing the solvent to evaporate overnight on petri plates, the solution was collected (Figure 11). The whole thing occurred at a comfortable 25 degrees Celsius. Researchers removed the polymer sheet and analyzed it, as shown in Figure 12 and Figure 13. The procedure adopted for synthesizing PMMA films containing the rest of the inorganic fillers was the same as mentioned above.

Figure 12 and Figure 13 show that the color intensity of PMMA films containing organo-clay-based CV/IC nano pigments was directly proportional to the loading of CV/IC onto the organoclays and in the polymer matrix (Table 3 and Table 4). As discussed in Section 3.1 and Section 3.2, the uptake of CV/IC was maximum in the case of OBent therefore, the color intensity of PMMA films containing OBent-based CV/IC nano pigments is maximum. The color intensity of PMMA films increased up to a point (150 mg nano pigments/5 g PMMA). In all cases, it was found that more than 150 mg nano pigment/5 g PMMA turned the films translucent, and the pigment particles became visible to the naked eye.

#### 3.7.3. Mechanical Properties of Synthesized PMMA Films

The mechanical properties of PMMA films containing pristine and organo Mt, Bent, and Vt-based CV/IC nano pigments were studied as a function of tensile strength, Young’s modulus, and hardness (Table 5, Table 6 and Table 7). Improvement in the tensile strength and Young’s modulus of PMMA films after adding CV, pristine, and organo clay-based nano pigments indicates better resistance against tensile stress and better stiffness.

The tensile strength and Young’s modulus of pure PMMA films were compared with PMMA films containing pristine and organo clay-based CV nano pigments [29,30] (Table 5). The tensile strength and Young’s modulus of PMMA films containing organo Mt and Bent were higher than PMMA films containing pure PMMA films and PMMA films containing pristine Mt and Bent-based CV nano pigments. The most significant improvement in tensile strength (by a factor of 7.8 concerning pure PMMA films) was observed in PMMA films containing OMt-based CV nano pigments, which was the highest among all types of PMMA films. The tensile strength and Young’s modulus of pristine PMMA films were compared with those of PMMA films containing pristine and organo clay-based IC nano pigments (Table 6). The tensile strength of PMMA films containing organo Mt, Bent, and Vt-based IC nano pigments increased by a factor of 5.18, 5.37, and 3.54 which was higher than PMMA films containing pristine Mt, Bent, and Vt-based IC nano pigments. An enhancement in Young’s modulus of PMMA films containing organo Mt, Bent, and Vt-based IC nano pigments was found to be high than PMMA films containing pristine Mt, Bent, and Vt-based IC nano pigments due to the hydrophobic nature of organo Mt, Bent, and Vt [31,32,33,34].

Hardness value indicates the effect of inorganic fillers on the resistance of polymer film deformation. The hardness of PMMA polymer films was found to increase after adding the filler. A great enhancement was observed after the addition of the organo clays-based CV/IC nano pigments, which can be further explained based on the hydrophobic nature of organoclays resulting in their better compatibility. No significant change was observed in tensile strength, Young’s modulus, and hardness of PMMA films containing Vt and OVt-based CV/IC nano pigments due to negligible ion exchange abilities. Enhancement in the mechanical properties of PMMA films containing organo clay-based nano pigments was higher, as organoclays filled the free spaces of the polymer chains, thus acting as reinforcement agents.

#### 3.7.4. Light Fastness Studies of Synthesized PMMA Films

The light fastness study describes the stability of dyes upon light exposure to a high level of UV radiation. The light fastness study of PMMA films containing CV was compared with PMMA film containing OMt-based CV nano pigments. Table 8 shows that PMMA film containing OMt-based CV nano pigments displayed a brighter color even after exposure to UV radiation. As the value of ΔL* was found to be more negative, indicated the darkness of the polymer films. The positive value of Δa* indicated the redness, and the negative value of Δb* indicated that the PMMA film shifted toward blue. The value of ΔE for PMMA films containing OMt-based CV nano pigment was less (12.25) than PMMA film containing pure CV (ΔE = 19.71), suggesting higher stability against UV radiation. The great enhancement in the color fastness of polymer films containing OMt-based CV nano pigment is due to their better compatibility with PMMA matrixes.

## 4. Conclusions

In the present work, the interaction of cationic as well as anionic dyes with the pristine clay minerals/clay (Mt, Vt, and Bent) and organoclays (OMt, OBent, and OVt) was investigated as a function of the pH of the dye solution, contact time, and initial concentration of dye. During the extraction experiments, it was found that the adsorption efficiencies of pristine clay minerals were higher for the cationic dye. Crystal violet and efficiencies of organoclays were higher for the anionic one, Indigo Carmine. Crystal Violet dye was found in the interlayer of pristine clay minerals and on their surface. In contrast, Indigo Carmine dye was found only on the surface of pristine/organo clays, as confirmed by XRD and Zeta potential data. During the interaction of the dye with pristine clay minerals and their organo forms, the colored solid materials obtained were further used as value-added products (clay-based nano pigment) for polymer matrix (PMMA) to form colored polymeric films instead of creating second-generation-waste materials. Due to the hydrophobic nature of organoclays, organo clay-based pigments dispersed homogenously in the polymer matrixes giving more intense color. The mechanical properties of PMMA films containing organo clays (OMt, OBent, and OVt) and CV/IC-based organo clays (OMt, OBent, and OVt) were found to be higher than those of PMMA films containing pristine clays (Mt, Bent, and Vt) and CV/IC-based pristine clays (Mt, Bent, and Vt) due to the better compatibility of organoclays with the PMMA polymer matrix. Therefore, it can be concluded that PMMA films containing organoclay and organo clays-based nano pigments showed better results compared to PMMA films containing pristine and pristine clay-based nano pigments.

## Figures and Tables

**Figure 1 micromachines-14-01087-f001:**
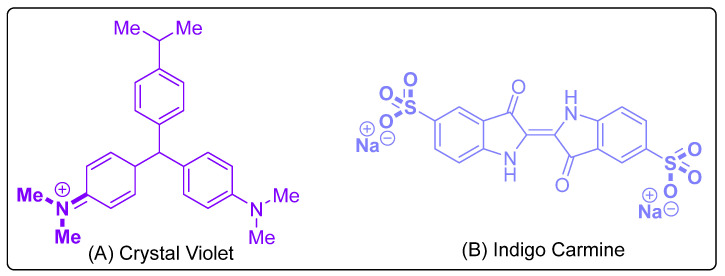
Structure of (**A**) Crystal Violet, (**B**) Indigo Carmine.

**Figure 2 micromachines-14-01087-f002:**
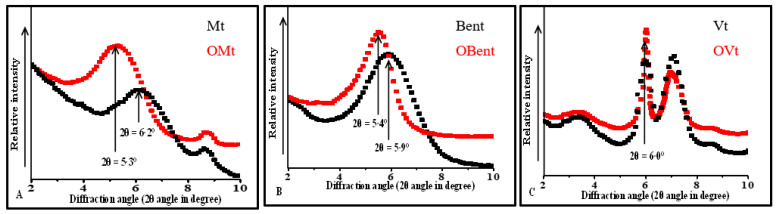
XRD pattern of clay before and after modification. (**A**) Mt. (**B**) Bent. (**C**) Vt.

**Figure 3 micromachines-14-01087-f003:**
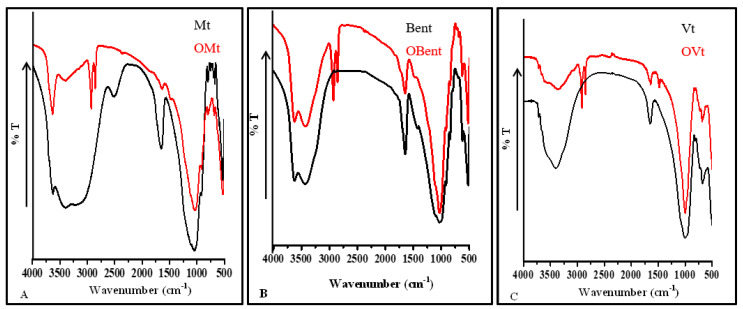
FT-IR spectra of clay before and after modification. (**A**) Mt. (**B**) Bent. (**C**) Vt.

**Figure 4 micromachines-14-01087-f004:**
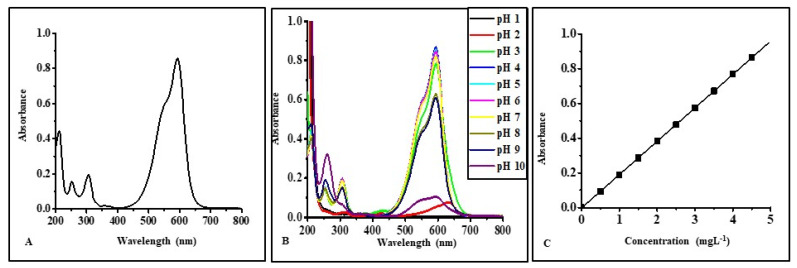
(**A**) Absorption spectrum of an aqueous solution of CV (4 mg L^−1^) at neutral pH (5.5), (**B**) effect of pH on UV-Visible spectrum of an aqueous solution of CV, (**C**) absorbance as a function of the concentration of CV in aqueous media at pH 4.

**Figure 5 micromachines-14-01087-f005:**
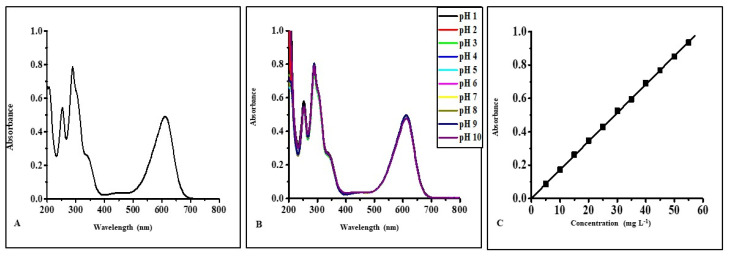
(**A**) Electronic absorption spectrum of an aqueous solution of IC (20 mg L^−1^) at neutral pH (4.2), (**B**) effect of pH on UV-Visible spectrum of an aqueous solution of IC, (**C**) absorbance as a function of the concentration of IC in aqueous media at pH 4.

**Figure 6 micromachines-14-01087-f006:**
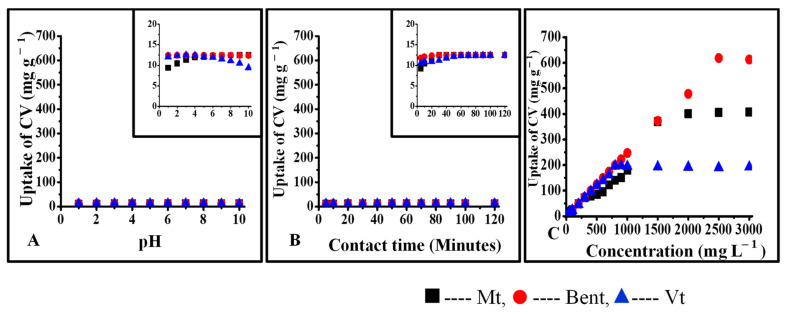
Interaction of CV with pristine clay: (**A**) interaction of CV as a function of pH of the solution, (**B**) interaction of CV as a function of contact time, (**C**) interaction of CV as a function of initial concentration.

**Figure 7 micromachines-14-01087-f007:**
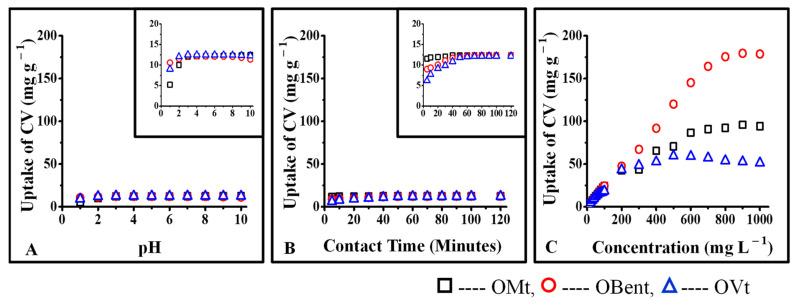
Interaction of CV with organoclays: (**A**) interaction of CV as a function of pH of the solution, (**B**) interaction of CV as a function of contact time, (**C**) interaction of CV as a function of initial concentration.

**Figure 8 micromachines-14-01087-f008:**
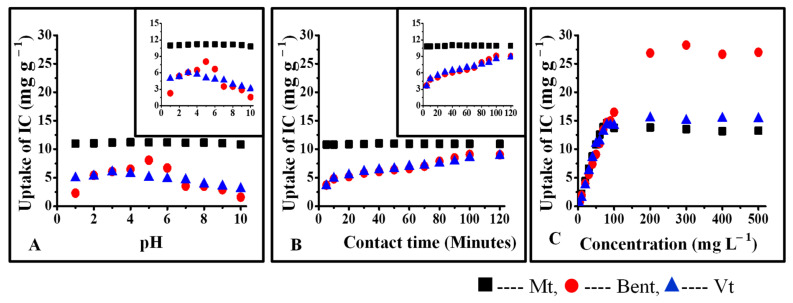
Interaction of IC with pristine clays: (**A**) interaction of CV as a function of pH of the solution, (**B**) interaction of CV as a function of contact time, (**C**) interaction of CV as a function of initial concentration.

**Figure 9 micromachines-14-01087-f009:**
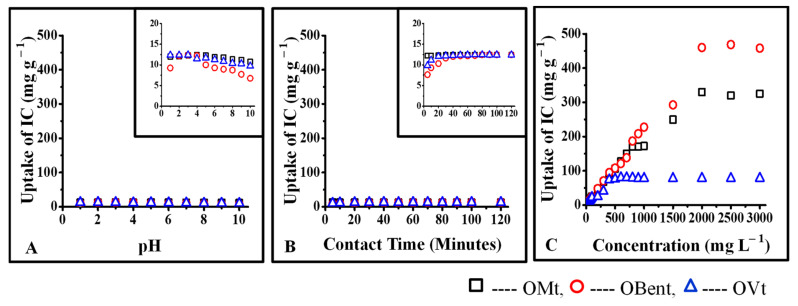
Interaction of IC with organoclays: (**A**) interaction of IC as a function of pH of the solution, (**B**) interaction of IC as a function of contact time, (**C**) interaction of IC as a function of initial concentration.

**Figure 10 micromachines-14-01087-f010:**
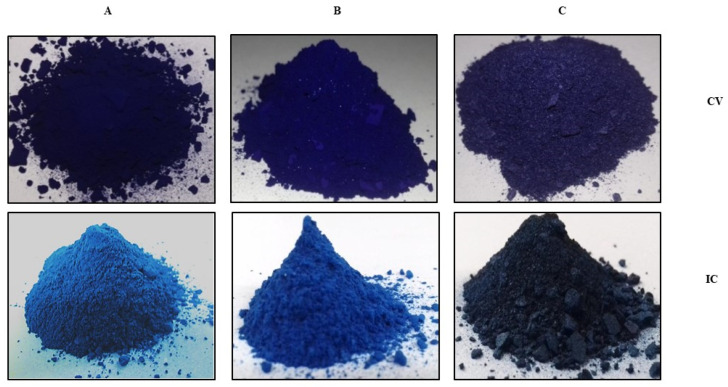
Clay based nano pigments: (**A**) Mt based nano pigment dry powder, (**B**) Bent based nano pigment dry powder, (**C**) Vt based nano pigment dry powder.

**Figure 11 micromachines-14-01087-f011:**
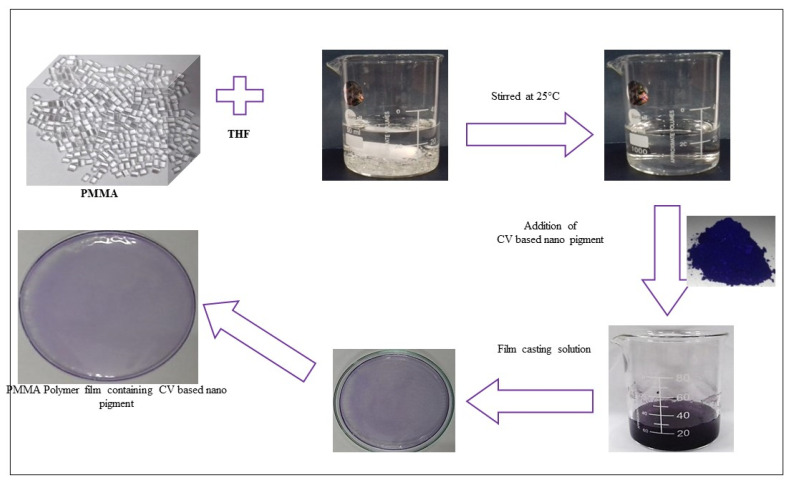
Flowchart for the synthesis of PMMA film.

**Figure 12 micromachines-14-01087-f012:**
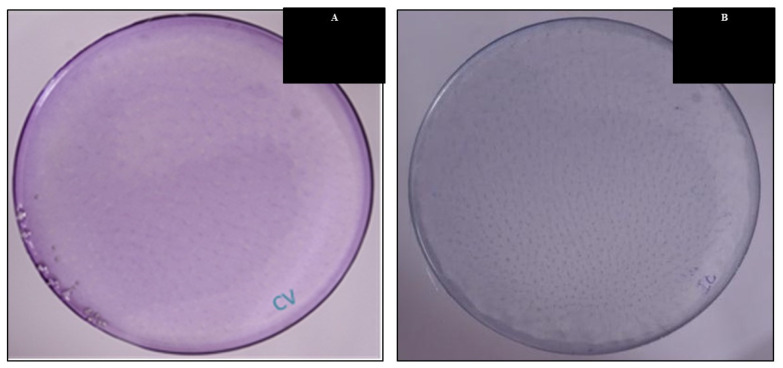
Pristine clay-based nano pigments dispersed PMMA films: (**A**) pristine clay-based CV nano pigment, (**B**) pristine clay-based IC nano pigment.

**Figure 13 micromachines-14-01087-f013:**
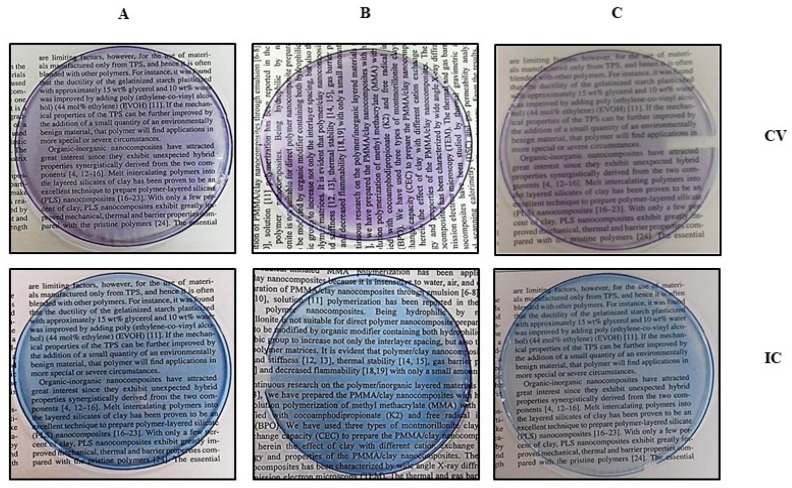
Organoclay-based nano pigments dispersed PMMA films: (**A**) Mt-based nano pigment dispersed polymer film, (**B**) Bent-based nano pigment dispersed polymer film, (**C**) Vt-based nano pigment dispersed polymer films.

**Table 1 micromachines-14-01087-t001:** Uptake of CV by pristine and organoclays and their characteristic properties.

Clay	Sample	CV Uptake mg g^−1^	Zeta Potential mV	Basal Spacing (d) Å	Particle Size nm
Montmorillonite	Mt	-	−17.60	14.45	119.0
	Pristine Mt + CV	407	−9.42	17.00	234.6
	OMt	-	−17.20	17.00	151.1
	OMt + CV	96	−10.40	14.12	181.4
Bentonite	Bent	-	−31.10	15.09	158.0
	Pristine Bent + CV	618	+35.50	32.21	245.0
	OBent	-	−24.50	16.38	251.0
	OBent + CV	180	−11.20	16.00	233.6
Vermiculite	Vt	-	−47.00	14.80	145.0
	Pristine Vt + CV	197	−14.60	14.61	236.6
	OVt	-	−19.10	15.10	995.4
	OVt + CV	59	−3.85	14.52	190.0

**Table 2 micromachines-14-01087-t002:** Uptake of IC by pristine and organoclays and their characteristic properties.

Clay	Sample	IC Uptake mg g^−1^	Zeta Potential mV	Basal Spacing (d) Å	Particle Size nm
Montmorillonite	Pristine Mt	-	−17.60	14.45	119.0
	Pristine Mt + IC	14.6	−11.40	14.58	140.6
	OMt	-	+29.30	17.50	174.0
	OMt + IC	330.0	−2.61	17.50	225.4
Bentonite	Pristine Bent	-	−31.10	15.09	158.0
	Pristine Bent + IC	28.0	−29.30	15.24	190.0
	OBent	-	+35.70	17.20	524.0
	OBent + IC	468.0	−13.40	17.00	300.6
Vermiculite	Pristine Vt	-	−47.00	14.80	145.0
	Pristine Vt + IC	15.5	−48.60	14.90	336.6
	OVt	-	−19.10	15.20	1062.0
	OVt + IC	80.0	−21.20	15.00	327.0

**Table 3 micromachines-14-01087-t003:** Loading of CV in PMMA film containing organo clay-based CV nano pigments.

S. N.	Adsorbent	Loading of CV (mg/g)		
		Organo Clay	PMMA	CV in PMMA Film (mg/cm^2^)
1.	OMt	96	2.28	0.0315
2.	OBent	180	5.40	0.0746
3.	OVt	59	1.77	0.0244

**Table 4 micromachines-14-01087-t004:** Loading of IC in PMMA film containing organo clay-based IC nano Pigments.

S. No.	Adsorbent	Loading of IC (mg/g)		
		Organo Clay	PMMA	IC in PMMA Film (mg/cm^2^)
1.	OMt	172	5.16	0.0713
2.	OBent	230	6.90	0.0953
3.	OVt	80	2.40	0.0331

**Table 5 micromachines-14-01087-t005:** Effect of pristine and organo clay-based CV nano pigments on the tensile strength and Young’s modulus of PMMA films.

S. No.	Sample Code	Stress (Pascal) (Average Value)	Strain (mm/mm) (Average Value)	Tensile Strength (MPa)	Standard Deviation	Young’s Modulus (MPa)	Standard Deviation
1.	PMMA	4.68	0.058	1.32	2.996	160.92	246.5651894
2.	PMMA + CV	9.68	0.025	2.85		354.10	
3.	PMMA + MtCV	25.14	0.040	7.82		731.12	
4.	PMMA + OMtCV	34.83	0.059	10.32		818.02	
5.	PMMA + BentCV	16.98	0.018	4.99		774.13	
6.	PMMA + OBentCV	21.45	0.032	5.48		875.15	
7.	PMMA + VtCV	25.20	0.042	7.98		637.45	
8.	PMMA + OVtCV	26.52	0.045	7.98		683.88	

**Table 6 micromachines-14-01087-t006:** Effect of pristine and organo clay-based IC nano pigments on the tensile strength and Young’s modulus of PMMA films.

S. No.	Sample Code	Stress (Pascal) (Average)	Strain (mm/mm) (Average)	Tensile Strength (MPa)	Standard Deviation	Young’s Modulus (MPa)	Standard Deviation
1.	PMMA	4.68	0.058	1.32	2.072	160.92	232.6868583
2.	PMMA + IC	10.15	0.035	3.02		355.28	
3.	PMMA + MtIC	12.01	0.028	3.45		490.25	
4.	PMMA + OMtIC	25.52	0.046	6.85		754.25	
5.	PMMA + BentIC	25.26	0.040	6.45		575.02	
6.	PMMA + OBentIC	25.45	0.051	7.10		920.25	
7.	PMMA + VtIC	13.48	0.018	4.28		602.12	
8.	PMMA + OVtIC	16.97	0.028	4.68		625.12	

**Table 7 micromachines-14-01087-t007:** Effect of pristine and organoclay-based IC nano pigments on the hardness of PMMA films.

S. No.	Sample Code	Hardness	S. N.	Sample Code	Hardness
1.	PMMA	79	-	-	-
2.	PMMA + CV	82	8.	PMMA + IC	82
3.	PMMA + MtCV	85	9.	PMMA + MtIC	83
4.	PMMA + OMtCV	91	10.	PMMA + OMtIC	91
5.	PMMA + BentCV	85	11.	PMMA + BentIC	85
6.	PMMA + OBentCV	89	12.	PMMA + OBentIC	89
7.	PMMA + VtCV	87	13.	PMMA + VtIC	83
8.	PMMA + OVtCV	87	14.	PMMA + OVtIC	85

**Table 8 micromachines-14-01087-t008:** Effect of CV and OMt-based-CV nano pigments on PMMA films.

Sample Code	ΔL*	Δa*	Δb*	ΔE
PMMA + pure CV	−6.73	−18.47	1.33	19.71
PMMA + OMt-based CV nano pigments	−11.68	3.69	−0.18	12.25

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
