# Peer review of "Synthesis of Nano Pigments Using Clay Minerals and Organic Dyes and Their Application as Colorants in Polymer Matrix"

_micromachines, 2023, doi:10.3390/mi14051087_

Round 1

Reviewer 1 Report

 It's a very interesting work, but it needs some revisions before being further processed.

1)     Page 3, line 93: ‘Before bed’ was mentioned in the manuscript, what does that mean, it is recommended to use professional terms.

2)     In the introduction section, why study? The reference for support your the aim is weak. Please cite the relative references.

3)     Zeta potentials of CV and IC was more helpful to understand the reason of the Zeta potentials changes of samples, and it is suggested to add them in the table 1 and 2.

4)     The higher adsorption efficiency of primary clay minerals on cationic dyes could be explained by the electrostatic interaction, but how to understand that modified clay minerals increase the efficiency of anions instead.

5)     In the section of Introduction and Materials, abbreviations for the three clay minerals (Montmorillonite, Bentonite, and Vermiculite) should be indicated when they first appear.

6)     In the cases of IC/pristine clays and IC/Organoclays, firstly, the surface charge changes obviously rather than “neither the surface charge changed much after exposure to IC”. In addition, another possible explanation for the small change in basal spacing is that the IC molecule is small in size and that interlayer insertion of clay minerals in a suitable manner may not cause a significant change in, which is recommended to be confirmed.

7)     It is recommended that the letters on Figure 10 were resized to fit the figure. In addition, ΔE and chrominance parameter (ΔL, Δb, Δa) should be in italics.

8)     From Figure 11, organoclay-based nano pigments do not evenly dispersed in PMMA films, which may cause large errors in the testing of mechanical properties. Therefore, the tensile strength and Young's modulus of each films should be measured at least five times to take the average value and add the standard deviation in the table.

9)     The whole manuscript should be thoroughly checked to correct mistakes in tenses and grammar, and taken care to standardize the format of the references.

Author Response

Please find the attached document for point-to-point responses.

Reviewer 2 Report

This paper reports the interactions between organic dyes (CV and IC) and clay minerals, and between organic dyes and organoclays, aiming to develop a novel methodology for synthesizing clay-based hybrid pigments without creating second-generation waste materials. However, there are some problems needing to be corrected or clarified. Moderate revision is necessary before acceptance. Detailed comments are given below.

1. This paper is awful organized, including the logic and language. I suggest to re-organize it carefully.

2. I should remind the authors that bentonite is not a clay mineral. Bentonite is a claystone where montmorillonite is the key component. Therefore, comparing the performance of montmorillonite and bentonite is unnecessary.

3. The XRD patterns of clay samples, organoclays and even the pigments should be provided. Also, the key features of these clay minerals, such as purity, chemical compositions, and cation-exchange capacity (CEC) should also be supplied.

4. Line 44-46: “No reports show the formation of nano pigments through the interaction of organic dyes with clay minerals by modulating various parameters” . This statement is not the truth.

5. Line 47-48: As I see it, montmorillonite and vermiculite are not real nanomaterials except for their exfoliation.

6. Line 49-53: Clay-dye hybrid pigments, from ancient Maya blue to modern synthetic ones, have been studied more. However, I did not find related relevant literature. There, I strongly suggest reviewing and citing the previous work carefully. Here I just give some examples below:

-Guillermin, D., Debroise, T., Trigueiro, P., de Viguerie, L., Rigaud, B., Morlet-Savary, F., Balme, S., Janot, J.-M., Tielens, F., Michot, L., 2018. New pigments based on carminic acid and smectites: A molecular investigation. Dyes Pigments 160, 971-982.

-Lima, L.C.B., Silva, F.C., Silva-Filho, E.C., Fonseca, M.G., Zhuang, G., Jaber, M., 2020. Saponite-anthocyanin derivatives: The role of organoclays in pigment photostability. Appl. Clay Sci. 191, 105604.

-Zhuang, G., Li, L., Li, M., Yuan, P., 2022. Influences of micropores and water molecules in the palygorskite structure on the color and stability of Maya blue pigment. Micropor. Mesopor. Mater. 330, 111615.

-Li, L., Zhuang, G., Li, M., Yuan, P., Deng, L., Guo, H., 2022. Influence of indigo-hydroxyl interactions on the properties of sepiolite-based Maya blue pigment. Dyes Pigments 200, 110138.

-Zhuang, G., Rodrigues, F., Zhang, Z., Fonseca, M.G., Walter, P., Jaber, M., 2019. Dressing protective clothing: stabilizing alizarin/halloysite hybrid pigment and beyond. Dyes Pigments 166, 32-41.

-Zhuang, G., Jaber, M., Rodrigues, F., Rigaud, B., Walter, P., Zhang, Z., 2019. A new durable pigment with hydrophobic surface based on natural nanotubes and indigo: Interactions and stability. J. Colloid Interf. Sci. 552, 204-217.

-Grazia, C., Buti, D., Amat, A., Rosi, F., Romani, A., Domenici, D., Sgamellotti, A., Miliani, C., 2020. Shades of blue: non-invasive spectroscopic investigations of Maya blue pigments. From laboratory mock-ups to Mesoamerican codices. Herit. Sci. 8, 1-20.

-Giustetto, R., 2019. Mayan Inspired Nanocomposite Materials: an Overview. Acta Phys. Pol., A 135, 1123-1125.

-Dong, J., Zhang, J., 2019. Maya Blue Pigments Derived From Clay Minerals, Nanomaterials from Clay Minerals. Elsevier, pp. 627-661.

…….

7. L71-77: Is this paragraph the main text?

8. L93: I don’t understand “Before bed”.

9. L294-315: This section is suggested to be placed in the Materials and Methods part

Author Response

(The authors gave the same response as above.)

Reviewer 3 Report

please check the attachment

Author Response

(The authors gave the same response as above.)

Round 2

Reviewer 1 Report

 The suggestions were inserted in the text. The quality manuscript was improved. Now, my suggestion is that this manuscript can be accepted to be published in this Journal.